**Data Availability Statement:** All relevant data are within the manuscript and its Supporting information files.

**Funding:** J.R. Uriarte acknowledges financial support from the Basque Government and the

# A behavioural model of minority language shift: Theory and empirical evidence

**José Ramón Uriarte** [1]◉*, **Stefan Sperlich** [2]◉

**1** Departamento de Fundamentos del Análisis Económico, Facultad de Ciencias Económnicas y Empresariales, University of the Basque Country, Bilbao, Basque Country, Spain, **2** Geneva School of Economics and Management, Université de Genève, Genève, Switzerland

◉ These authors contributed equally to this work.
* jr.uriarte@ehu.eus

## Abstract

Natural languages with their speech communities tend to compete for speakers, very much like firms compete for market shares. As a result, some languages suffer a shifting pressure which might lead them to their extinction. This work studies the dynamics of language shift in the context of modern bilingual societies like the Basque Country, Ireland and Wales. They all have two official languages, linguistically distant: *A*, spoken by all, and *B*, spoken by a bilingual minority. They also have a bilingual education system that ensures a steady flow of new bilinguals. However, a decay in the use of *B* is observed, signalling that shift processes are at work. To investigate this apparent paradox, we use a novel approach in the literature of language competition. We build a behavioural game model with which bilinguals choose either language *A* or *B* for each interaction. Thus, they play repeatedly the game. We present a theorem predicting that under reasonable assumptions, any given population of bilinguals will converge into a linguistic convention, namely into an evolutionary stable equilibrium of the game, that always embeds a proportion of bilinguals shifting to *A*. We validate this result by means of an empirical version of the model, showing that the predictions fit well the observed data of street use of Basque and daily use of Irish and Welsh.

## Introduction

In societies with two languages, a majoritarian one *A*, and a minoritarian one *B*, the process of abandoning the use of *B* in favour of *A* is called minority language shift. Due to its importance for the cultural diversity, this is a widely studied topic by sociolinguists (e.g., [1–3]), and more recently also by physicists and mathematicians. To our knowledge, the first formal model of language shift is the influential work of [4] (some extensions of that model are [5–7]; see also the survey of [8] and the references therein). Reaction-diffusion models have also been proposed to study the dynamics of language shift over time and space; see [9–11]. Further, [12] observe that none of the formal models provides information on local changes in language use. Hence, they propose a microscopic scale model that can follow successfully the changes in language use on a detailed spatial scale.

Spanish Ministerio de Economía y Competitividad (PID2019-106146GB-I00)]. The funders had no role in the study design, data collection and analysis, decision to publish, or preparation of the manuscript.

**Competing interests:** The authors have declared that no competing interests exist.

However, while some of these formal models may fit well the data, they explain less well why bilingual individuals behave one way or the other. The reason is that no adaptive decision-makers nor behavioural assumptions are introduced in those models. Instead, linguistic interactions are modelled for instance as a Gaussian function, c.f. [12] who consider a model of physical diffusion. Along this line, there are models that describe language competition and shift as travelling waves and wave fronts (e.g., [10, 11]). Although the inspiration for modelling based on physics is legitimate, the problem is that since bilingual individuals are not described as decision makers, language shift cannot be observed as the outcome of their language choice behaviour. Therefore, the introduction of behavioural assumptions to understand minority language shift remains a challenge. To complement the literature on formal models of language shift, our approach is to start with a theoretical behavioural model, and then counter-check it with observational data.

We will adopt the Council of Europe's view ([13]) that one should look at the bilinguals' language use behaviour, and concentrate our study on three European, contemporary, modern bilingual societies: the Basque Country, Ireland and Wales. They all have two official languages, linguistically distant, the majoritarian *A*, and the minoritarian *B*. The chance of speaking *B* in these advanced bilingual societies is sensitive to the forces of modernization. In particular, the mobility, both social and geographical, of the work force (composed of bilingual and monolingual individuals) and the growing of metropolitan cities shape anonymous interactions, which means that people's linguistic type (bilingual or monolingual) becomes private information. Further, since bilinguals and monolinguals live in the same areas and have jobs in the same workplaces, language contact between *A* and *B* ([14, 15]) is now tighter than in past periods. New laws and abundant resources are used in those societies in favour of language *B*. Political changes were made that led to the recognition of their minority languages as official ones: the Law of Normalization of Basque's Use of 1982; the Welsh Language Measure of 2011; and the recognition by the Constitution of the Republic of Ireland in 1937 that Irish is the first official language. Hence, to the bilingual speakers of these societies, the *A* and *B* languages have the same perceived value or status. Therefore, the minority language prestige, which is considered to be a relevant parameter for [4], becomes meaningless to understand language shift in the societies under consideration. As in all modern societies, the main transmission channel of society's cultural traits is the educational system, where pupils receive a bilingual education in the two official languages; see [16–18]. Thus, a continuous flow of potentially bilingual individuals is guaranteed. However, the data of Basque, Irish and Welsh show that despite the existing proportions of bilinguals, the use of these languages is lagging behind. This apparent paradox signals that language shift processes are at work.

The societies under study have the following three relevant features: the languages *A* and *B* are linguistically distant. There are no geographical or any other type of barriers between bilinguals and monolinguals (i.e. monolinguals are those who only speak *A*, and, due to language distance, do not understand *B*); that is, bilinguals and monolinguals are in close contact. Furthermore, people's linguistic type is private information. Under these conditions, the language decisions taken by bilinguals have an interactive nature. That is, since the chosen language will serve the purpose of communication in interactions where bilinguals and monolinguals might meet, the choice of *B* made by a bilingual might not be the actual language that will be used in the interaction. Note that this type of interactive decisions hardly occurs in a monolingual society, i.e. a society where only one language is spoken. Game theory and, in particular our proposed game which we call the Language Use Game model (LUG, henceforth), is a natural framework to study interactive language decisions, allowing us to introduce behavioural assumptions. In our model, bilingual individuals choose the language for each interaction through the play of such LUG. A bilingual would choose *B* when she plays the strategy *Reveal*

that you are bilingual. The bilingual will choose language *A* when she plays the strategy *Hide* that you are bilingual. Since monolinguals do not choose language, the LUG is a game played only by the (sub)population of bilinguals. Further, since there is uncertainty about the linguistic type of the interactive partners, we assume that the LUG is a game played under imperfect information. To overcome this imperfection, a bilingual who chooses the strategy *Reveal* is informing about her bilingual nature to the interactive partner, so that *B* will be used if both happened to be bilingual. Then they will get a higher utility for using their preferred language *B*. But a *Reveal* player may suffer a utility loss when she is forced to switch to *A* when the interactive partner happens to be monolingual. However, when a bilingual chooses the strategy *Hide* she is reinforcing the imperfect information, making the use of *B* less likely. In game-theoretic terminology, one could say that we model this repeated play of the LUG by means of a so-called replicator dynamics equation (RD). This RD is derived for social contexts from imitation and trial-and-error-learning processes, i.e. in our case the players repeatedly play the LUG replicating their *Reveal* / *Hide* strategy (see also further below). Accordingly, the RD model is the right framework, though the presentation of our model below can also be followed without knowing RD; you may consult [19, 20] for details. Since we want to trace down the local (i.e. municipal) changes in the use of *B*, we assume that the LUG is played at every municipality quoted in the sociolinguistic surveys. These are periodically made to know the state of *B* (either Basque, Irish or Welsh). To understand the language shift occurring at a local level, we seek to predict the use of *B* associated to the municipal population of bilinguals observed in the specific year of each survey.

We present a theorem saying that under some reasonable assumptions, any given municipal population of bilinguals will converge into a linguistic convention that always embeds a proportion of bilinguals shifting to *A*. In other words, the repeated play of the LUG makes any population of bilinguals converge into an equilibrium that has the nature of a linguistic convention. This convention takes the form of a partition of the population in two groups: those who play *Reveal* and those who play *Hide*. Members of the former group speak in *B* with any other bilingual. But when *Hide* players meet, then they will use language *A*. In terms of language dynamics, we show that the linguistic conventions have strong stability properties. To empirically validate this theoretical finding, we build a statistical version of the model, and show that the equilibrium predictions fit well the observed local changes in the street use of Basque, and daily use of Irish and Welsh, respectively. We conclude that our model is empirically valid to understand minority language shift dynamics in the considered modern bilingual societies. How would previous formal models of language shift dynamics (quoted above) perform in the bilingual societies considered here? They may break down when they do not handle the imperfect information of bilingual types, with bilinguals having difficulties to recognize each other. Further, since bilinguals and monolinguals form tight mixed populations across the country, there are no speech communities of *A* segmented geographically or economically from the speech communities of *B*. Accordingly, reaction-diffusion models are not appropriate for the societies under consideration. Therefore we see our contribution as a complement to the literature on formal models of language shift. Finally, our model gives a (probably partial) view of the formal issues one should look at to tackle Fishman's question "Why is it so hard to save a threatened language" [1].

## Materials and methods

### Benchmark bilingual societies

We consider economically advanced bilingual societies having two official languages, *A* and *B*, linguistically distant in the spirit of [21]. We assume that there are essentially two linguistic

**Table 1. Evolution of percentages of bilinguals (*α*), street use (*KE*) and daily use (*DU*).**

| Basque | | | Irish | | | Welsh | | |
|--------|-----|------|------|------|------|------|------|------|
| Year | *α* | *KE* | Year | *α* | *DU* | Year | *α* | *DU* |
| 1993 | 22.3 | 11.8 | | | | | | |
| 1996 | 24.4 | 13.0 | 1996 | 41.1 | 10.16 | | | |
| 2001 | 25.4 | 13.3 | 2002 | 41.9 | 09.05 | | | |
| 2006 | 25.7 | 13.7 | 2006 | 40.8 | 02.10 | 2005 | 20.8 | 13.0 |
| 2011 | 27.0 | 13.2 | 2011 | 40.6 | 02.13 | | | |
| 2016 | 28.4 | 12.6 | | | | 2014 | 19.0 | 13.0 |

These percentages were not always measured in the same year; and surveys were executed over longer periods. The indicated year refers to the mid-term.

groups: the monolingual speakers, who speak only *A*, and a minority of bilingual speakers who speak *A* and *B* with similar skills. Let $\alpha \in (0, 1)$ denote the proportion of bilingual speakers over the total population of the specific society, and $1 - \alpha$ the proportion of monolinguals, see Table 1. We do not take into account the boundary numbers 0 and 1, which are hardly observed in the data used in the present work anyway. The first assumption is the following:

**A.1**. *Languages and Speakers*: *The languages with official status, A and B, are linguistically distant so that successful communication is only possible in a single language, either A or B. Further, the proportion of bilinguals in the total population of the country is smaller than the proportion of monolinguals*: $\alpha < 1 - \alpha, \alpha \in (0, 1)$.

Passive bilinguals, i.e. those who understand *B* but cannot speak it, are not considered (or can be treated as monolinguals as discussed in the S1 Appendix. When a monolingual interacts with a bilingual, they necessarily use *A*. Hence, *B* is spoken only when two bilinguals meet, and at least one of them signals the desire to speak in *B*. Therefore, language choice is not trivial.

Some examples of societies satisfying **A.1** are the Basque Country, Ireland and Wales. Basque is a pre-indoeuropean language, spoken in the Autonomous Community of Euskadi (ACE) (Spain), Navarre (Spain) and in the Pays Basque (France). It is in contact with the two Romance languages, Spanish (Castellano) and French. Irish and Welsh, two Celtic languages, are in contact with the Germanic language English. These are competitive bilingual societies, both in the economic and the linguistic domain, with sufficient resources to implement well articulated language policies to protect and transmit *B*, mainly through the educational system. They are a kind of benchmark in the set of societies with threatened languages satisfying **A.1**. If minority languages are shifted here, this will be even more so in the less developed multilingual societies contemplated in [1]. Note that because of **A.1** our model may not apply to study the issues of closely related languages, such as Catalan and Castellano. The use of a minority language can be measured in different ways. For instance, since 2006, the Census of Population in Ireland contains a questionnaire about the ability to speak Irish for aged three and over, and the frequency of use (daily, weekly, less often, never) outside the Education system. The Census of Wales contains a similar questionnaire with respect to Welsh. In Wales, two language use surveys were carried out covering the periods 2004–06 and 2013–15, with a self-completion questionnaire to be returned by post. The information gathered from those sources serve to estimate the daily use (*DU*) of Irish and Welsh (data sources for Irish and Welsh are in S1 Appendix). A quite different procedure is followed to measure the *street use* of Basque -*kale erabilera* in Basque (*KE*). Field surveyors collect random observations in the streets and public places of municipalities where Basque is spoken, by listening and recording the language of conversation between the observed subjects. The information gathering procedure is

anonymous since there is no contact between surveyors and observed subjects, neither before nor after the recording. For more details, see S1 Appendix.

In Table 1 are shown the aggregates obtained in the mentioned surveys. They show the proportion of bilinguals, $\alpha$, the street use of Basque, $KE$, and the daily use of Irish and Welsh, $DU$, in different years. However, for the empirical study, we will use the data on the disaggregated level, namely municipality and local authority level, as they exhibit the joint variation of $\alpha_i$ and language use, $KE_i$ and $DU_i$ ($i$ denoting a municipality or local authority). This is extremely important because in our context, aggregated data can be quite misleading. Take the following example of looking at only two localities (denoted 1 and 2), a small one with $\alpha_1 = 0.8$ and $KE_1 = 0.5$, and a five times larger locality with $\alpha_2 = 0.2$ and $KE_2 = 0.02$. For their aggregates we get $\alpha = 0.3$ with $KE = 0.1$. Now, supposing that all conversations were the result of a random match (people match independently of their language skills), if bilinguals communicated among themselves always in $B$, we had $KE_1 = \alpha_1^2 = 0.64$ and $KE_2 = \alpha_2^2 = 0.04$; i.e., in each locality there would be a much more frequent street use of $B$ than the observed ones such that one could conclude a language shift. In contrast, if only looking at the aggregates, we could expect at most $KE = \alpha^2 = 0.09$ which is even less than observed. Consequently, the aggregates would insinuate a frequent use of $B$, whereas the disaggregated data could reveal a language shift.

**Remark 1**: The sources of data to elaborate the linguistic surveys of these three languages are the localities: municipalities for Basque, and local authorities for Irish and Welsh. Our model will be applied on those local data. Access to all the data used in the present work is free, see S1 Appendix.

## An example: The rules of the road game

Before we continue, the following example will provide you with an intuitive idea of the notions of (strict) Nash equilibrium and Evolutionary Stable Strategy Equilibrium as an emerging Convention, terminology used in the next section. Let us consider the game of which side of the road to drive on, whether left side or right side [22]. The game is played by two oncoming vehicles. Since drivers want to avoid accidents, they will coordinate regarding the side of the road (i.e. a coordination game). Drivers have no incentives to deviate from coordinating on the same side because they are strictly better off than being involved in an accident. This is the basis for the notion of strict Nash equilibrium. Hence, this coordination game has two strict Nash equilibria: both on their left side (L,L) or both on their right (R,R) side. These equilibria are two conventions. But which of the two equilibria will be chosen? This is the same as asking which of the two conventions, (L,L) or (R,R), will be adopted by the society? To answer this question, we should view how the game evolves historically by the repetition of the interaction. We can think of the game being played by a large population. Each time a different pair of drivers will play the game and may remember how the interaction was solved. As the game is repeated over and over by many people, one particular way of solving the interactions, say (R,R), takes prominence in the population, driving out the other one, (L,L). This process of emerging the convention is by accumulation of precedents (in the game theory literature one often speaks of 'imitation'). In our view, this mechanism might explain how a linguistic convention is built by the bilinguals. A convention may become established through a different mechanism. Initially, when traffic was scarce, some area of the country might adopt, by local custom, say, the (L,L) convention, while other areas of the country adopt the (R,R) convention. As traffic grew, the evolution of those conventions became affected by sudden changes in the law that regulated the traffic. Finally, the society decided politically to establish one convention. Note that in terms of welfare, both conventions are equally good. What the established convention does is to solve the indeterminacy. That is, we may define a

convention that has emerged historically as an equilibrium that everybody expects in interactions (i.e. expectations and behaviour are finally in an equilibrium). This could also have more than one equilibrium, as in the road game. Such a convention that has emerged and been adopted by the society, is called an Evolutionary Stable Strategy Equilibrium of the game. For our Language Use Game below, it can be shown (see S1 Appendix) that bilinguals converge into an equilibrium in which they do not all play a strategy that would maximize the use of $B$.

## The language use game

We assume that this the LUG is played in every locality of the country where $B$ is spoken. For each year of the linguistic survey of $B$, a locality $i$ is characterized by the reported number of bilingual speakers, $N_i$, the proportion of bilinguals with respect to the total population of $i$, $\alpha_i$, and the reported local street use of Basque, $KE_i$, or the local daily use of Irish and Welsh, $DU_i$, respectively. To simplify notation, we do not use indices to refer to each of the three languages since the base model is the same for all languages and localities. We use the linguistic surveys to understand how a decrease in the use of $B$ ca be explained albeit there is an education system ensuring a continuous flow of new speakers of $B$. We proceed as follows. We choose the year of the survey of one of the considered $B$ languages, and take as given the reported $\alpha_i$ of locality $i$. Then we seek to predict with the LUG model the use of $B$ in locality $i$ for that year. We use the same procedure for every locality reported in the survey. Thus, for the chosen year we obtain a function that relates each reported $\alpha_i$ with its corresponding predicted use of $B$. Then we compare the predictions with the actual data of $B$ use in that year. We proceed in the same manner for every linguistic year of each of the three $B$ languages. Notice that 'predicted by' or 'expected along' our model can be used synonymously in this context. A detailed description and derivation of the LUG can be found in S1 Appendix. Here we concentrate on the main ideas, starting with the assumptions.

A feature of modern societies is the mobility, both social and geographical, of the work force, and the growing of metropolitan cities. The spread of bilinguals in those urban contexts implies that often they interact without perfect knowledge about the others' language skills and preferences.

**A.2**. *Linguistic Imperfect Information*: *Let $i$ be any locality of a country where one of the mentioned B languages is spoken. We assume that the linguistic type (bilingual or monolingual) of individuals interacting in $i$ is private information. We also assume that bilinguals, both from $i$ and from elsewhere in the country know the proportion $\alpha_i$ reported in the linguistic surveys.*

We may assume, at least for bilinguals, common knowledge about the proportions of bilinguals, $\alpha_i$, because when the linguistic surveys are published, their content is discussed in the media, commented by the concerned people, and politically debated. As it is common when one is testing empirically a theory, we must adapt **A.2** to the reality of actual conversations (see below the statistical version of the model). We can think that each bilingual individual is adapted to the sociolinguistic context of her locality $i$, where language $B$ is known and used by a percentage of the population. That linguistic landscape will shape a specific adaptation level or reference point for the bilingual inhabitants of $i$ (as it happens with non-sensory attributes, such as wealth or health, and with sensory ones, such as temperature) cf. [23]. Thus, the locality where the bilingual inhabits becomes her linguistic reference point. It is natural to assume that the bilinguals have the aspiration of $B$ becoming a non-endangered language. Beyond that, however, they know $B$ will not reach language $A$'s full normalization state. We can think that there is an increasing and concave (but almost flat) aspiration function $S$ that assigns to each $\alpha_i$ the aspiration proportion $\alpha_i^*$, which the bilinguals from $i$ think it would make $B$ a non-endangered language, fulfilling $\alpha_i < \alpha_i^* < 1$. Given the flat curvature of $S$, all aspirations are

grouped into a similarity interval [24]. That is, for all $i$, $\alpha_i^*$ has no perceptible differences with the country's aspiration set by the linguistic authority, denoted $\alpha^*$. Therefore we will use $\alpha^*$ instead of $\alpha_i^*$, fulfilling $\alpha_i^* < \alpha^* < 1$ for all $i$. Notice that in a choice situation, the *aspiration* represents the most desired alternative, available or not. However, it is well known, experimentally and theoretically, that unavailable choices have an important influence on the decision behaviour of agents, known as *the aspiration effect* [25].

The payoffs in terms of 'experienced utilities' are assumed to represent net communication benefits ([26–28]). Let $m(\alpha_i)$ be the payoff that a bilingual gets when interacting in language $B$ at locality $i$. Let $n - c(\alpha_i) > 0$ be the payoff to a bilingual in locality $i$ who, having decided to use $B$, is matched to a monolingual but due to **A.1** is forced to switch to $A$. Thus, $c(\alpha_i)$ is the bilingual's frustration cost. Since information is imperfect (**A.2**), a bilingual may choose voluntarily language $A$, obtaining $n > 0$. This $n$ is also the payoff for monolinguals from using the majority language $A$. Therefore, $n$ is a constant. We make the following assumptions:

**A.3**. *Aspirations, Linguistic Preferences and Payoffs: Let us consider the set of all localities i where one of the mentioned B languages is spoken, and the set of proportions of speakers of B of each locality, $\alpha_i$, reported in the linguistic survey of a given year. The payoff function for speaking B and the frustration cost for being forced to switch, $m(\alpha_i)$ and $c(\alpha_i)$, satisfy the following properties*:

- *$m(\alpha_i)$ and $c(\alpha_i)$ are strictly decreasing functions in $\alpha_i \in (0, \alpha^*)$, approaching n and zero, respectively, as $\alpha_i$ approaches the country's aspiration level $\alpha^*$.*

- *Given $\alpha_i < \alpha^*$, $m(\alpha_i) > n$, where $n > c(\alpha_i) > 0$ and $c(\alpha_i) < (m(\alpha_i) - n)\frac{\alpha_i}{(1-\alpha_i)} =: b(\alpha_i)$ (weighted benefit).*

Note that $m(\alpha_i) > n$ shows that the preference intensity for using $B$ decreases as $\alpha_i$ increases (i.e., when we observe localities with higher proportion of bilinguals).

**Remark 2**: The existence of a bilingual educational system produces a continuous flow of potential bilinguals that changes the $\alpha_i$. Due to mobility, the increase can be quite unequal over the localities. Those changes are mainly due to factors outside the model, related to political attitudes, cultural identity, and the commitment of transmitting $B$ to future generations. This means that the argument of communicative benefits [27] as a driving force to invest in the learning of a language, hardly applies here.

The LUG is a game in strategic form because the purpose is to investigate bilinguals' interactive language choices. If we were to model a conversation, then it would be better described by a so-called extensive form game; see [29]. Specifically, imagine how the game is played at any locality $i$ (where either Basque, Irish or Welsh is spoken) endowed with a proportion $\alpha_i \in (0, \alpha^*)$ in a given year. We assume that bilinguals choose the language at the beginning of an interaction, under the assumptions **A.1**-**A.3**. They do it by choosing one of the following pure strategies of language use:

**R**: *Reveal always that you are bilingual, so that you will speak B whenever you meet a bilingual.*

**H**: *Hide that you are bilingual, and reveal it only when matched with an R-player. That is, speak A, and switch to B only when you are addressed in B.*

When an $R$ player meets a bilingual, their common (bilingual) trait is revealed, and language $B$ will be used. In this event, both bilinguals get the maximum payoff $m(\alpha_i)$. In the event of meeting a monolingual, by assumption **A.1** the $R$ player is forced to switch to $A$ and obtains $n - c(\alpha_i)$. However, a monolingual might feel uncomfortable when meeting an $R$ player and being forced to confess her ignorance of $B$. The $H$ player avoids that discomfort by choosing $A$ as speaker (the interactant who starts the conversation). As a respondent, she answers in the

**Fig 1. The language use game.** The matrix on the left describes the state where two bilinguals meet. The combination of pairs of strategies, **R** and **H**, gives rise to cells where a pair of payoffs and the resulting language are shown. The column on the right describes the state where a Bilingual meets a Monolingual. The spoken language will be *A*, and the payoff to the bilingual depends on the strategy chosen.

language used by the speaker, either *A* or *B*. This way the *H* player avoids the cost $c(\alpha_i)$ but gets $m(\alpha_i)$ (only) when matched with an *R* player. The essence of *H* is that she reinforces the imperfect information **A.2**, behaving as a *monolingual in disguise*, unless she is discovered by an *R* player. Clearly, *H* is less effort demanding than *R*. Note that neither the mentioned strategies nor being speaker or respondent are conditioned by a specific player position (row or column) in the LUG. The pure strategies of the LUG describe two frequent behaviours observed in minority bilingual populations. The inspiration for *R* is the *militant* of the minority language (for the *militants* of Basque, who are gathered in *Euskaraldia*, and those of Welsh, mostly gathered in the *Welsh Language Society*, see S1 Appendix). Whereas *H* represents a more conventional behaviour, who tend to think it is not polite to address in *B* to unknown interlocutors, and easily falling into language shift The LUG, described in Fig 1, is a game played, at each locality *i*, only by bilinguals since monolinguals do not choose language. There are two states of nature: two bilinguals meet (Bilingual), or a bilingual meets a monolingual (Monolingual). At the start of the interaction, the bilingual, by **A.2**, is uncertain about the state, but expects to meet another bilingual with probability $\alpha_i$, and expects to meet a monolingual with probability $1 - \alpha_i$. When the Bilingual state is realized, depending on the pair of strategies, either language *A* or *B* is spoken and a pair of payoffs is obtained. When the Monolingual state is realized, the bilingual will speak *A*, and get $n - c(\alpha_i)$ if *R* is chosen, or payoff $n > 0$ if *H* is chosen. Hence, strategy (i.e. language) choices are made according to expected payoffs. The matrix of expected payoffs is given in Figure, see also S1 Appendix.

Assumptions **A.1** and **A.2** capture the conditions under which bilinguals must make their language choices. Given those constraints, bilinguals are led to make frequent language choices to satisfy their linguistic preferences (**A.3**). Hence, we assume that the LUG is played continuously over time, at each locality *i*, by the local bilingual population $N_i$. Now consider a given survey year. As noted above, we seek to predict the use of *B* in each locality *i*, given its proportion $\alpha_i$. Hence, we may imagine that in every locality *i*, pairs of individuals from this locality are repeatedly drawn at random to play the LUG. We assume that the resulting language use population dynamics is modelled by the standard replicator dynamics (RD) attached to the LUG (S1 Appendix). The replicators are the pure strategies *R* and *H*. In the standard RD setting, there is at any time *t* a language use population state, $(p_i(t), 1 - p_i(t))$, where $p_i(t) = N_{iR}(t)/N_i$ represents the fraction of bilinguals of locality *i* who play pure strategy *R* at time *t*, and $1 - p_i(t)$ the fraction of those who play *H* at *t* (we will abbreviate $p_i(t)$ to $p_i$ in what follows). Since we have a population of bilinguals who can play only two pure strategies, the RD consists of a single ordinary differential equation that describes the growth rate of $p_i$ (the growth rate of the fraction who plays *H* will move in the opposite direction; adding up both growth rates must always be equal to 1). The next result makes a theoretical prediction about the use of language

*B*, in a randomly chosen locality *i*. We make the mild assumption that at the starting point of the replicator equation, there are some bilinguals playing *R*; that is, $p_i \in (0, 1)$ at $t = 0$. The theorem shows that under the assumptions **A.1**-**A.3**, any given local population of bilinguals involved in a continuous play of the LUG will converge into a linguistic convention that will always contain a proportion of that population shifting to language *A*. Remark 3, below, describes the convention in terms of the strategic context of the LUG.

**Theorem 1** *Let B denote any of the minority languages under consideration, and i a randomly chosen locality in which the linguistic survey (of the year under scrutiny) reports a proportion of speakers of B, $\alpha_i \in (0, \alpha^*)$. Then under the assumptions **A.1**-**A.3**, and $p_i \in (0, 1)$ at $t = 0$:*
*(a) The ESS equilibrium: given the observed $\alpha_i$, the replicator dynamics (i.e., the language use population dynamics) converges into $p_i^* \in (0, 1)$, the unique Nash equilibrium which is an evolutionary stable strategy (ESS) of the LUG played at i:*

$$p_i^* = p^*(\alpha_i) = 1 - \frac{(1 - \alpha_i)c(\alpha_i)}{\alpha_i(m(\alpha_i) - n)} = \frac{\alpha_i(m(\alpha_i) - n) - (1 - \alpha_i)c(\alpha_i)}{\alpha_i(m(\alpha_i) - n)} \tag{1}$$

*where $n > 0$ is a given constant. There are two additional rest points for the replicator equation, 0 and 1, both unstable. (b) Language B shift in equilibrium: if $p_i^* = p^*(\alpha_i)$ is the ESS equilibrium reached at i, then $(1 - p_i^*)(1 - p_i^*)$ is the probability of a random matching of two bilinguals who play strategy H and use language A in the interaction; i.e., the probability of language shift in equilibrium at locality i, in the year under consideration. And $p_i^* \times p_i^* + p_i^*(1 - p_i^*) + (1 - p_i^*)p_i^*$ is the probability of all random matches in which an R player participates and B is used at locality i, for that year.*

Proof: Part (a) We calculate first the LUG´s matrix of expected payoffs (see S1 Appendix) and then we build the associated replicator dynamics equation, to obtain:

$$\dot{p}_i = p_i(1 - p_i)[\alpha_i(m(\alpha_i) - n)(1 - p_i) - c(\alpha_i)(1 - \alpha_i)]$$

By **A.3**, $p_i^* = p^*(\alpha_i) \in (0, 1)$. From the replicator equation, it can be seen that $p_i^*$ is a global attractor in (0, 1), and that both 0 and 1 are unstable rest points (see S1 Appendix). Part (b) follows from the ESS equilibrium.

**The ESS equilibrium function**: Assume now that the LUG is played (under the same set of assumptions of the Theorem) not just in one but in every locality *i* where the language *B* we are dealing with is spoken. Then each locality's language use population dynamics will converge into an ESS in the interval (0, 1), as in the Theorem. We may, equivalently, say that the ESS equilibrium of the Theorem, $p_i^* = p^*(\alpha_i)$, becomes a function, having as domain the set of all local $\alpha_i$ reported in the language survey of the year under consideration. Thus, we have the following Corollary of the Theorem.

**Corollary 1** (*The language B shift equilibrium function*): *Using* Eq (1), *and under the assumptions of the Theorem, we build a function $p^*(\alpha_i)$ as follows: To each value of $\alpha_i \in (0, \alpha^*)$, reported in the survey of the year under consideration, $p^*(\alpha_i)$ will assign its corresponding local ESS, $p_i^*$. Therefore, this function predicts, for each given $\alpha_i$, the language B shift of equilibrium at locality i, for that year.*

**Remark 3**: In social and economic contexts, an ESS equilibrium is thought of as a convention [30]. Thus, in the context of the LUG, each $p_i^*$ is a linguistic convention that takes the form of a partition of the bilingual population of each locality *i*, $N_i$, in two groups: $N_i p_i^*$ and $N_i(1 - p_i^*)$. Members of the former group speak *B* with any other bilingual, and members of the latter speak *A* between them. Like any other convention, it becomes a self-enforcing mechanism of language coordination, with the unintended consequence of a fraction of bilinguals shifting to language *A*.

## The statistic predictive models of minority language use

The predictive models are based on the expected use of *B* in the ESS equilibrium (developed in S1 Appendix). The predictive street use of Basque and the predictive daily use of Irish and Welsh are modelled in different ways. We start with the latter since it is somewhat easier to model. Let $p_i^* = p^*(\alpha_i)$ be the corresponding ESS equilibrium reached in that locality (as shown in the above Theorem).

*Predictive model of Daily Use of Irish and Welsh*:

$$PDU(\alpha_i) = c_0 \alpha_i p_i^*$$

for an unknown positive constant $c_0$.

The reasoning behind this is that bilinguals who play strategy *R* will answer (in the language survey or Census) that they use *B* every day, and that almost all who play strategy *H* will answer that they do not use *B* every day. This suggests $c_0 \approx 1$, which, however, is not needed in our study, see Eq (5) below.

For modelling the street use of *B* at locality *i*, it is natural to assume that, in the sample population, conversations in *B* occur both in random and non-random matches. Suppose we treat all street conversations at *i* like random matches between two people. Then the chance of observing the use of *B* would be the probability to record a militant (i.e., the prototype of an *R* player of the LUG model, as mentioned above) matched with another bilingual, independently of whether the latter is a militant or not. That probability, in the ESS equilibrium, would be: $(\alpha_i^2 p_i^{*2}) + \alpha_i^2 p_i^*(1 - p_i^*) = \alpha_i^2 p_i^*$, where $\alpha_i p_i^*$ is the chance to observe a militant, and $\alpha_i$ the chance that the other person is bilingual. Thus, $\alpha_i^2 p_i^*$ has to be multiplied by 2 because the order, (militant + bilingual) or (bilingual + militant), does not matter for the language use. But, then, since the matches of two militants are counted twice, they have to be subtracted once. You get the same formula by looking at all combinations via the binomial formula:

$$(\alpha_i^2 p_i^{*2}) + 2\alpha_i^2 p_i^*(1 - p_i^*) = 2\alpha_i^2 p_i^* - \alpha_i^2 p_i^{*2} \tag{2}$$

However, a street language use survey will get also the records of nonrandom matches, which will, most likely, increase the observed use of *B* due to the linguistic preferences of bilinguals; in particular, the militants' preferences. Moreover, militants, as minority language advocates, are more likely (i.e., more frequently) to know or recognize each other, meet and talk. This suggests that Eq (2) will experience two kind of changes caused by the non-random matches. First, the percentage of conversations in *B* increases as a whole; and second, for the above mentioned reasons, the fraction of matches with two militants involved will increase. All this transforms Eq (2) into the following model:

*Predictive model of Street Use of Basque*:

$$PKE(\alpha_i) = c_1\left(2\alpha_i^2 p_i^* - c_2 \alpha_i^2 p_i^{*2}\right)$$

with unknown constants $c_1 > 0$, $c_2 \leq 1$.

Clearly, for $c_1 = c_2 = 1$ we have the same proportion as in the unrealistic situation of only recording random matches. One may argue that $c_1 \approx 1$ assuming that the change is mainly due to the increased matches of militants. For the same reason $c_2$ is likely to be much smaller than 1. We tried different values and found an excellent data fit for $c_2 = 0$, which implies that the fraction of matches between militants doubles, see Eq (4) below.

## Empirical specification

Notice that, given the year of the linguistic survey, $PKE(\alpha_i)$ and $PDU(\alpha_i)$ are both functions that relate each locality's $\alpha_i$ with the predicted use of $B$ in each locality's ESS equilibrium. Now, we look for the statistical versions of $PKE(\alpha_i)$ and $PDU(\alpha_i)$ to provide predictions of the LUG model along the observed data of $KE_i$ and $DU_i$. To this end, we first need to build the empirical version of the equilibrium function $p_i^* = p^*(\alpha_i)$, by means of the empirical specification of the functions entering into Eq (1) (for a detailed derivation of the empirical form see the S1 Appendix). Plugging all the specifications into Eq (1), we obtain

$$p_i^* = p^*(\alpha_i) = \beta_1(1 - \alpha_i)(\alpha_i^{\beta_3} - \alpha_i)^{\beta_2 - 1} \ . \tag{3}$$

Substituting the empirical version of $p^*(\alpha_i)$ in $PKE(\alpha_i)$ and in $PDU(\alpha_i)$, using $c_2 = 0$, we get the empirical predictive models of $B$ use:

$$PKE(\alpha_i) = 2c_1\beta_1\alpha_i^2(1 - \alpha_i)(\alpha_i^{\beta_3} - \alpha_i)^{\beta_2 - 1} = \tilde{\beta}_1\alpha_i^2(1 - \alpha_i)(\alpha_i^{\beta_3} - \alpha_i)^{\beta_2 - 1}, \tag{4}$$

$$PDU(\alpha_i) = c_0\beta_1\alpha_i(1 - \alpha_i)(\alpha_i^{\beta_3} - \alpha_i)^{\beta_2 - 1} = \check{\beta}_1\alpha_i(1 - \alpha_i)(\alpha_i^{\beta_3} - \alpha_i)^{\beta_2 - 1}, \tag{5}$$

We introduce here $\tilde{\beta}_1 = 2c_1\beta_1$ and $\check{\beta}_1 = c_0\beta_1$ because $\beta_1$ cannot be separately identified in either of the models. Since we do not need to identify it for our study, we can use this simplified notation.

**Remark 4**: For each year of the linguistic survey, the empirical versions of $PKE(\alpha_i)$ and $PDU(\alpha_i)$ have one observational variable, the $\alpha_i$ of each locality, and three regression parameters: $\beta_2, \beta_3$ and either $\tilde{\beta}_1$ or $\check{\beta}_1$. The estimates of these parameters are presented in S1 Appendix. For the model to make sense, we work with the restrictions $\tilde{\beta}_1, \check{\beta}_1 > 0$ and $\beta_3 \in (0, 1)$.

## Empirical results

The presentation of results concentrates on the predictive functions. In Fig 2 the estimates of the empirical equilibrium models, the functions $PKE(\alpha_i)$ and $PDU(\alpha_i)$, are summarized. The individual estimates are depicted in S1 Appendix.

Not surprisingly, each empirical equilibrium model, $PKE(\alpha_i)$ for Basque, and $PDU(\alpha_i)$ for Irish and Welsh, gives rise to an increasing and convex relation between $\alpha_i$, and the predicted use of $B$ in the ESS equilibrium at $i$. It could be argued that intuition might suggest that the predictive functions should be increasing and convex. However, these two properties alone cannot explain the variations observed in the data. There is still plenty of room for model misspecification. For this reason, we tested nonparametrically the model based estimates. We first calculated nonparametric regression functions for all years and languages, and then constructed 90% confidence bands for our model-based estimates using bootstrap. Details are given in S1 Appendix, together with all the resulting figures. All model-based estimates come very close to the nonparametric data fits. Moreover, most of the time the confidence bands include the nonparametric fit. This supports empirically our theory, and therefore, the LUG model. Indeed, other theory models could be constructed leading to similar results. This, however, is in the nature of empirical analysis and holds for all empirical studies.

Let us look at the development of $\alpha_i$, $KE_i$ and $DU_i$ over time. The box-plots (left panel) in Fig 2 illustrate the development of the distributions over the years. Recall that we are looking at all combinations $(\alpha_{ti}, KE_{ti})$ and $(\alpha_{ti}, DU_{ti})$, without weighting them by the population size of municipality $i$. We see that all years exhibit a huge dispersion for $\alpha_i$ and $KE_i$ (less so for $DU_i$) with no stabilization of the distributions of these indicators. The panel on the right of Fig 2

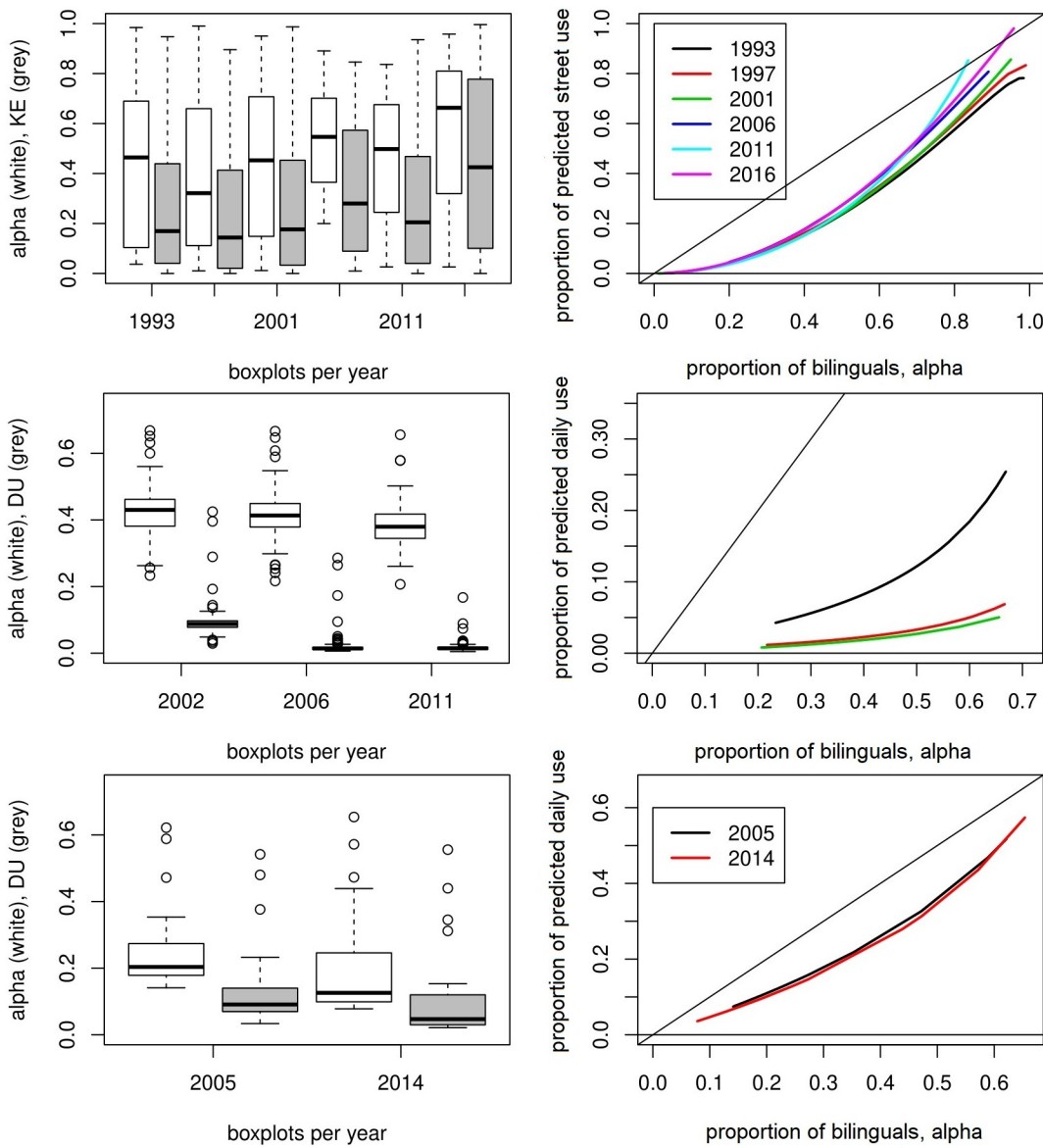

**Fig 2. Development of α and use of Basque (upper panel), Irish (centre) and Welsh (lower panel) over time.** For each year, $PKE(\alpha_i)$ associates to each local $\alpha_i$ the predicted local street use of Basque in equilibrium; and so does the $PDU(\alpha_i)$ for Irish and Welsh. On the left we see box-plots showing the development of $\alpha$ (white) and $KE$, respectively $DU$ distributions (grey) over years.

shows the development of $PKE(\alpha_i)$ and $PDU(\alpha_i)$ over time. The tendency for Irish (ys 2002, 2006, 2011) is clear: its use went down dramatically, already starting from a low level. The $PDU(\alpha_i)$ captures the drop of Irish use between 2002 and 2006 (see Table 1). That fall could be due to the fact that only since 2006 the daily use of Irish was clearly defined as used outside the education system. Basque and Welsh have a more intensive use than Irish. For Welsh we have two observations in time, contained in the surveys WLUS 2004–06 and 2013–15 about the use of Welsh. Thus, we show two $PDU(\alpha_i)$ that show no significant changes, capturing the stability observed in the WLUS data for that decade (yet, see the discrepancies between the Censuses and the WLUS surveys in S1 Appendix). The Basque language follows a more intriguing path

(than Irish and Welsh). The data for Basque covering the period from 1993 to 2016 show that the number of municipalities with small $\alpha_i$ and/or small values of $KE_i$ has diminished. This might explain the bilinguals' aspirations, and why they feel that the country's proportion of bilinguals, $\alpha$, has to increase much more to become a non-endangered and almost normalized language ($\beta_3$ is converging to zero from above; as in Wales, but less so in Ireland; see S1 Appendix. We first briefly comment on some observed changes in the data, and then show how the model captures and explains those changes. Consider municipalities with a small percentage of bilinguals ($\alpha_i < 20\%$) such as those that compose the metropolitan area of Bilbao (35 municipalities and 1,037,847 inhabitants, almost half of the total population of the Basque Autonomous Community in 2016). The changes of $\alpha_i$ in those localities reached its maximum in 2011, and remain at a similar level for the next years. $KE_i$, increased too, with some municipalities reaching its maximum in 2001, others in 2006 or 2011. Once the maximum is reached, essentially all municipalities show a decrease of the $KE_i$ in the following years. Thus, in some localities, the increase in $\alpha_i$ coincides with a decrease in $KE_i$, like in Bilbao, where $\alpha_i$ starts at 9.63% in 1997 and reaches a maximum of 24.30% in 2011. Whereas $KE_i$ starts at 4.52% in 1997 and goes down to 2.5% in 2016. Consider now municipalities where the majority of the population is bilingual, say, with $\alpha_i \geq 60\%$, like those in the coast of the province of Biscay. There, roughly speaking, the percentage of bilinguals is kept constant during the period 1993–2016. At the same time, again the $KE_i$ increases initially, but decreases in later periods, almost everywhere. Since the use of Basque was historically on a high level, the decrease could be quite pronounced, like in Bermeo, where $KE_i$ reached its maximum, 60.35%, in 2006, but then fell to 33.6% in 2016.

The changes in the local data are well captured by our predictive model. For low values of $\alpha_i$, the $PKE(\alpha_i)$ function of 1993, 1997, 2001, 2011 and 2016, do not show significant upward movements toward the diagonal since the net changes in $KE_i$ are very small. But, as mentioned above, they show that the number of localities with very low $\alpha_i$ and $KE_i$ have diminished. For high values of $\alpha_i$, above 50%, the model shows that the curves move toward the diagonal. That is, if we choose an $\alpha_i$ (above 50%), and keep it constant, we see that the corresponding $KE_i$ increases, at different rates, from 1993 to 2011. However, this upward tendency is reversed in 2016. That year's $PKE(\alpha_i)$ curve is below the previous one of 2011, and almost overlapping with that of 2006 in communities with high $\alpha_i$.

How can we explain the decline in the use of $B$ in areas where bilinguals are historically a majority of the population and have been using $B$ as a habit? Our model proposes an explanation that hinges mostly on assumption **A.3**. Bilinguals will be adapted to those high proportions of knowledge and use of $B$, $\alpha_i$ and $KE_i$, that will shape their *reference point*. Then they might overestimate, by assuming that they are close to the country's aspiration, $\alpha^*$, with $B$ being out of danger. Then, by **A.3**, $m(\alpha_i)$ and $c(\alpha_i)$ would be close to their limit values, $n$ and zero, respectively. That is, bilinguals of those localities would not experience any perceptible utility benefit from using $B$, nor any utility loss from, occasionally, being forced to use $A$. Even in the presence of cultural identity loyalty, no perceived payoff differences between the two official languages means low preference intensity for $B$. This could lead bilinguals to develop indifference between the $A$ and $B$ languages. A direct consequence of language indifference is that bilinguals either code-switch frequently (alternating, words and expressions, between the $A$ and $B$ languages) or speak $A$ more often than $B$. However, the methodology followed to gather street use of Basque data does not count conversations displaying code-switching or words uttered in different languages (see S1 Appendix). The data indicate that for Basque such a change in the linguistic habits of bilinguals in high $\alpha$ contexts started around the year 2000, and much earlier for Irish.

## Discussion and conclusion

Many factors outside our model will influence the use of a minority language, such as globalization and international market integration; see [31, 32]. However, we have taken an internal view, drawing the attention to the bilinguals' interactive language decisions. Rather than extending theories of language shift based on models of physical diffusion (either as a Gaussian function or as a Fourier's law of heat conduction) explored in [12, 33], our approach proposes a behavioural game model. This framework allows to establish a functional relation between minority language shift and bilinguals' language choices. We show that the diffusion of language $B$ replacement, in each locality $i$ at a moment of time, is measured by the empirical specification of each locality's ESS equilibrium $p_i^*$, and the derived empirical models of predicted use of $B$, $PKE(\alpha_i)$ and $PDU(\alpha_i)$. Hence, our model is able to trace the local changes in language use.

Looking at the data, one wonders why some bilinguals would behave in a paradoxical way by shifting to $A$. Answers based on the prestige (i.e., the perceived status) of the minority language $B$, introduced by [4] and used in almost all the literature rooted in that work, would fail to understand this issue. As we mentioned above, the bilinguals of the considered societies wish that $B$ remain as an official language, and, further, have the aspiration of $B$ becoming a non-endangered language, fighting to minimize the impediments to its use. Similarly, travelling waves and linguistic fronts based on reaction-diffusion models are of little help in this respect since the bilinguals under consideration are spread all over the country in tight contact with monolinguals, forming a compact, non-segmented bilingual societies.

The answer given by our model would say: in high $\alpha_i$ contexts, bilinguals could develop language indifference between $A$ and $B$, giving rise to conversations plagued with code-switching between both languages (which could lead to a more intensive use of $A$, and a progressive shift). In low $\alpha_i$ contexts, the chooser of strategy $H$ minimizes potential frictions because it is supposed to be polite with monolinguals. But the choice of $H$ reinforces both the linguistic imperfect information and the use of language $A$ between bilinguals. Thus, in a situation of language contact with a powerful language $A$, the combination of language distance and imperfect linguistic information, could result in a behaviour embedding language indifference and politeness-led language choices. This, in turn, would lead bilinguals into a dynamics converging into linguistic conventions, as ESS equilibria, with relatively low proportions of $R$ players. Thus, a mixture of language choice conditions (specified by **A.1**-**A2**) and the derived bilinguals' choice behaviour act as the driving forces of the dynamics of language $B$ replacement. We dare to say that the bilinguals' behaviour is guided, as it is common in human language, by economizing attitudes ([34]), and the Zipf's principles of least effort: [28, 35].

The empirical result of this work is that the LUG is a solid model to understand minority language shift dynamics in the modern, benchmark, bilingual societies studied in the present work. The model provides knowledge of the formal relations between $\alpha_i$ and the predicted use of $B$, suggesting policy actions to control the shift process. Essentially, policy makers could address all the factors of Eq (1) in order to increase $p_i^*$ (i.e. language $B$ militants) in each locality $i$. Further, they could take measures to reduce the imperfect information about linguistic types. To our knowledge, the imperfect information on linguistic type that obstructs bilinguals to recognize each other, has been neglected in language economics, [36, 37], sociolinguistics [38], and by nearly all the mathematical models of language dynamics. Note that outside our behavioural game framework, the model of [33] can cope with lack of information of a different nature. Their Language Inheritance Principle II can estimate the inheritance rate parameter of the competing languages even when there is not sufficient data of speakers' language

choices. They predict then language dynamics by means of the Lotka-Volterra equations. Finally, our model suggests that linguistic politeness behaviour ([39]) is empirically relevant.

We are aware that a model is a simplification and can only explain a part of the observed dynamics. We considered various modifications, e.g., to allow for non-random matches or relaxing the assumption of imperfect information. While the prediction models get sometimes more complex, the empirical findings in the robustness checks did not change.

## Supporting information

**S1 Appendix. Appendix to the main document.**
(ZIP)

**S2 Appendix. All used data sets.**
(XLSX)

## Acknowledgments

J.R. Uriarte acknowledges the hospitality at both Humbolt-Universität Berlin and the University of Geneva. We thank the editor and the two reviewers whose comments helped us to improve the manuscript. Further, we thank Nagore Iriberri and Joel Sobel for their comments to an earlier version of this work. We are also grateful to the comments by Salvador Barberá and to Enrique Zuazúa for the endless and fruitful conversations on this topic. Finally, JRU wants to dedicate this work to I. Idiazabal for her comments and constant support, and to Maider Elixabete Uriarte Idiazabal, our morning sky.

## Author Contributions

**Conceptualization:** José Ramón Uriarte, Stefan Sperlich.

**Data curation:** José Ramón Uriarte, Stefan Sperlich.

**Formal analysis:** José Ramón Uriarte, Stefan Sperlich.

**Funding acquisition:** José Ramón Uriarte, Stefan Sperlich.

**Investigation:** José Ramón Uriarte, Stefan Sperlich.

**Methodology:** José Ramón Uriarte, Stefan Sperlich.

**Project administration:** José Ramón Uriarte, Stefan Sperlich.

**Resources:** José Ramón Uriarte, Stefan Sperlich.

**Software:** José Ramón Uriarte, Stefan Sperlich.

**Supervision:** José Ramón Uriarte, Stefan Sperlich.

**Validation:** José Ramón Uriarte, Stefan Sperlich.

**Visualization:** José Ramón Uriarte, Stefan Sperlich.

**Writing – original draft:** José Ramón Uriarte, Stefan Sperlich.

**Writing – review & editing:** José Ramón Uriarte, Stefan Sperlich.

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
