## [Decision Letter · Decision Letter 0]

15 Feb 2021

PONE-D-21-00751

A Behavioural Model of Minority Language Shift: Theory and Empirical Evidence

PLOS ONE

Dear Dr. Uriarte,

Thank you for submitting your manuscript to PLOS ONE. After careful consideration, we feel that it has merit but does not fully meet PLOS ONE’s publication criteria as it currently stands. Therefore, we invite you to submit a revised version of the manuscript that addresses the points raised during the review process.

We look forward to receiving your revised manuscript.

Kind regards,

Boyu Zhang, Ph.D

Academic Editor

PLOS ONE

Additional Editor Comments:

Both reviewers agree that this paper studied an interesting and important question. However, they feel that the mathematical model is too complicated and difficult to understand. Since target readers are linguists, the author may provide some detailed explainations for background knowledge, e.g., evolutionary game method, and discuss why this method is better than other existing methods. Furthermore, since the model has several parameters, the author should provide intuitions for why these parameters are necessary and how they work.

Journal Requirements:

Reviewers' comments:

Reviewer's Responses to Questions

**Comments to the Author**

1. Is the manuscript technically sound, and do the data support the conclusions?

Reviewer #1: Yes

Reviewer #2: Yes

2. Has the statistical analysis been performed appropriately and rigorously? 

Reviewer #1: Yes

Reviewer #2: I Don't Know

3. Have the authors made all data underlying the findings in their manuscript fully available?

Reviewer #1: Yes

Reviewer #2: Yes

4. Is the manuscript presented in an intelligible fashion and written in standard English?

Reviewer #1: No

Reviewer #2: Yes

5. Review Comments to the Author

Reviewer #1: This paper proposes language use game and uses Nash equilibrium theory and evolutionary stable solution to estimate the outcome of language shift in a community with a monolingual A and bilingual A & B. The authors briefly described the method and math behind and conducted experiments using theoretical setting and a setting supported by real data.

My general feeling is that the adoption of equilibrium based math in language shift modeling is a great innovation, though it requires deep math knowledge to understand, which may drive away linguists who are interested in the topic of language shift. To better understand the relevant concepts, it would be great that the authors provide some simple examples of ESS and background knowledge, before jumping directly to the complex math model. Keep in mind that the readers of your paper may not come from physics or math, but historical/contact linguistics, who may not have sufficient knowledge to understand your current writing, no understanding, no use.

In addition, why using such math, whether it is because it has some advantages that other approaches cannot provide, or this method can avoid some parameters that are hard to estimate from real data. The authors mentioned something similar only in the conclusion section (why other models cannot address the issues you raised), it would be great to move that part early in the introduction, and also, additional justification is needed.

Finally, given the parameters in the model, is it possible to conduct an analysis on the effect of major parameters, like alpha or alpha star, i found the current values are set pretty arbitrary.

minor issues:

line 23, delete ","

line 40, 'european' 'European'

the 2nd paragraph in introduction is too long, and incorporate many points, better to separate each point into one paragraph.

line 126: why street use of Basque is short for KE?

line 151: "could is it" -> 'is it'

line 470: citefortEtal2012 must be changed to reference

Reviewer #2: In my opinion this is an important attempt to understand linguistic behaviour of people in bilingual societies and may add to policy makers' decisions (and to the awareness of speakers of language B) when trying to save endangered languages.

My knowledge of game theory and statistics is not sufficient to judge the corresponding equations, but I feel they are O.K.

Some details:

Why not show data points of recorded observations in the figures in the main text?

Fig.2: I think the axis should read "fraction", not "%", same in figures in supp.info.

Alpha, not alpa.

Line 470: Fort and Perez-Losada

Supp.info:

p.2, Ireland: delete "As for 1996 this distinction is not at all clear we skipped this wave in the revised version of our paper."

p.9: "..where ??? and Nij(t)=Ni represents the share of bilinguals of locality i playing pure strategy j at t (j = R; H)."

p.12: (***)

6. PLOS authors have the option to publish the peer review history of their article (what does this mean?). If published, this will include your full peer review and any attached files.

Reviewer #1: No

Reviewer #2: No

---

## [Author Response · Author response to Decision Letter 0]

29 Mar 2021

Response to Editor: In an extensive letter, we have answered to every single comment raised by the Editor.

Responses to Reviewer 1 and 2 : Similarly, both reviewers´comments have been taken into account and have received the required attention and response in extensive letters.

---

## [Decision Letter · Decision Letter 1]

29 Apr 2021

PONE-D-21-00751R1

A Behavioural Model of Minority Language Shift: Theory and Empirical Evidence

PLOS ONE

Dear Dr. Uriarte,

Thank you for submitting your manuscript to PLOS ONE. After careful consideration, we feel that it has merit but does not fully meet PLOS ONE’s publication criteria as it currently stands. Therefore, we invite you to submit a revised version of the manuscript that addresses the points raised during the review process.

We look forward to receiving your revised manuscript.

Kind regards,

Boyu Zhang, Ph.D

Academic Editor

PLOS ONE

Additional Editor Comments (if provided):

Reviewer 1 is generally satisfied with the revision. However, reviewer 2 has some questions that need to be addressed. Please explain these questions in detail.

Reviewers' comments:

Reviewer's Responses to Questions

**Comments to the Author**

1. If the authors have adequately addressed your comments raised in a previous round of review and you feel that this manuscript is now acceptable for publication, you may indicate that here to bypass the “Comments to the Author” section, enter your conflict of interest statement in the “Confidential to Editor” section, and submit your "Accept" recommendation.

Reviewer #1: All comments have been addressed

Reviewer #2: All comments have been addressed

2. Is the manuscript technically sound, and do the data support the conclusions?

Reviewer #1: Yes

Reviewer #2: Yes

3. Has the statistical analysis been performed appropriately and rigorously? 

Reviewer #1: Yes

Reviewer #2: Yes

4. Have the authors made all data underlying the findings in their manuscript fully available?

Reviewer #1: Yes

Reviewer #2: Yes

5. Is the manuscript presented in an intelligible fashion and written in standard English?

Reviewer #1: Yes

Reviewer #2: Yes

6. Review Comments to the Author

Reviewer #1: some minor points to clarify. Zhang & Gong's work can also handle imperfect cases, like the ones with missing data or ongoing competition, your replies are not accurate. You may clarify this in the revised version.

Other than this, the other comments seem well addressed in the revised version.

Reviewer #2: Thank you for answers to my questions/comments and for the appealing example.

I still think that this is a highly interesting new approach to language shift touching on psychology and sociology, and that it certainly deserves publication. The way it is presented, however, is a real challenge to possible readers not familiar with game theory, and that could be the overwhelming majority.

Being until now not familiar with game theory I tried to learn the concepts and to check whether the conclusions of the paper appear plausible. I have to confess that still I can not follow all steps.

• On page 8 you write: “The replicators are the pure strategies R and H.” For non-specialists (e.g. me) at least a little surprising. How can…? But certainly correct (for game theorists), could you explain in detail?

I think that in particular Theorem 1 (page 9) could be presented in a way which permits better access to non-specialists.

Some ideas:

• Realize that the terminology might be repulsive for linguists et al. Could you try not to shock non-specialists with terminology like: “standard replicator dynamics”, “replicator dynamics equation” in other place just called “replicator equation”, “evolutionary stable state strategy ESS”, “Nash equilibrium”, “global attractor”, most of which presented in just 3 lines on page 9? Of course, by searching in Wikipedia also non-specialists can learn what all that means, but will they do so? And I doubt that you really need all of this terminology: it could be mentioned in footnotes for specialists who otherwise might be frustrated to miss it.

• Structure the explanation of the most important Theorem 1, i.e. in particular allow more space for deriving the replicator equation for the derivative of p_i. Possibly without making extensive use of game-theory terminology.

I believe equ. (1) is correct, but have to confess that I was not able to convince myself.

Typing errors and alike:

p.5 “…understand how is it possible a decay…”

Fig.2: I still believe that the labelling of both axes must be “proportion” (or fraction) and not %. Also Fig. 2 of SI.

SI p.8: perceive, not perceived.

SI p.13, line 7 from bottom: please replace g prime by g dot (twice).

SI p. 15: S1-S3 Tables, S1-S3 Figs. should read Tables S1-S3 and Figs. S1-S3, resp.

Terminology

Please define “private information”.

Is there a difference between “payoff” and “benefit”? On SI p.13 you even sometimes change from benefit to profit, that appears confusing. And how about “utility”, e.g. “perceptible utility gain” (SI p.7, third line from bottom) - what is difference from payoff or benefit? My confusion continues on SI p.8 where new term “preference intensity” appears.

SI 9: new term: “one-population replicator dynamics”. Why “one-population”?

SI p.10: “Hawk-Dove Game”�delete! If you love that term, please put into footnote.

SI p.11: why p_i^* ɛ(0, 0.293)?

SI p.12-13: what is difference E(KE) – P(KE), E(DU) – P(DU), resp.? Is “predicted” not the same as “expected”?

7. PLOS authors have the option to publish the peer review history of their article (what does this mean?). If published, this will include your full peer review and any attached files.

Reviewer #1: **Yes: **Tao Gong

Reviewer #2: No

---

## [Author Response · Author response to Decision Letter 1]

10 May 2021

All the comments by the Editor, Reviewers 1 and 2 have have been answered.

---

## [Decision Letter · Decision Letter 2]

17 May 2021

A Behavioural Model of Minority Language Shift: Theory and Empirical Evidence

PONE-D-21-00751R2

Dear Dr. Uriarte,

We’re pleased to inform you that your manuscript has been judged scientifically suitable for publication and will be formally accepted for publication once it meets all outstanding technical requirements.

Kind regards,

Boyu Zhang, Ph.D

Academic Editor

PLOS ONE

Additional Editor Comments (optional):

Congratulations! All revewers' comments have been addressed properly and they are satisfied with the current version.

Reviewers' comments:

Reviewer's Responses to Questions

**Comments to the Author**

1. If the authors have adequately addressed your comments raised in a previous round of review and you feel that this manuscript is now acceptable for publication, you may indicate that here to bypass the “Comments to the Author” section, enter your conflict of interest statement in the “Confidential to Editor” section, and submit your "Accept" recommendation.

Reviewer #2: All comments have been addressed

2. Is the manuscript technically sound, and do the data support the conclusions?

Reviewer #2: Yes

3. Has the statistical analysis been performed appropriately and rigorously? 

Reviewer #2: Yes

4. Have the authors made all data underlying the findings in their manuscript fully available?

Reviewer #2: Yes

5. Is the manuscript presented in an intelligible fashion and written in standard English?

Reviewer #2: Yes

6. Review Comments to the Author

Reviewer #2: Thanks für answering my questions, resp. misunderstandings, and explaining derivation which I had not understood before.

I hope that non-specialists will be able to follow the short presentation of theorem 1 (which now is "explained" by ONE more sentence).

Wish many interested followers!

Thank you indeed for your lectures on game theory and "eco-socio-psycho" issues.

7. PLOS authors have the option to publish the peer review history of their article (what does this mean?). If published, this will include your full peer review and any attached files.

Reviewer #2: No

---

## [Editor Report · Acceptance letter]

24 May 2021

PONE-D-21-00751R2 

A behavioural model of minority language shift: theory and empirical evidence 

Dear Dr. Uriarte:

I'm pleased to inform you that your manuscript has been deemed suitable for publication in PLOS ONE. Congratulations! Your manuscript is now with our production department. 

Kind regards, 

on behalf of

Dr. Boyu Zhang 

Academic Editor

PLOS ONE